# Mental Health Impact of Early Stages of the COVID-19 Pandemic on Individuals with Pre-Existing Mental Disorders: A Systematic Review of Longitudinal Research

**DOI:** 10.3390/ijerph20020948

**Published:** 2023-01-04

**Authors:** Angela M. Kunzler, Saskia Lindner, Nikolaus Röthke, Sarah K. Schäfer, Maria-Inti Metzendorf, Alexandra Sachkova, Roxana Müller-Eberstein, Carmen Klinger, Jacob Burns, Michaela Coenen, Klaus Lieb

**Affiliations:** 1Leibniz Institute for Resilience Research (LIR), 55122 Mainz, Germany; 2Institute for Evidence in Medicine, Medical Center—University of Freiburg, Faculty of Medicine, University of Freiburg, 79110 Freiburg, Germany; 3Department of Psychiatry and Psychotherapy, University Medical Center of the Johannes Gutenberg University Mainz, 55131 Mainz, Germany; 4Institute of General Practice (ifam), Medical Faculty, Heinrich Heine University Düsseldorf, 40225 Düsseldorf, Germany; 5Department of Anesthesiology, University Medical Center of the Georg August University Göttingen, 37075 Göttingen, Germany; 6Chair of Public Health and Health Services Research, Institute for Medical Information Processing, Biometry, and Epidemiology (IBE), LMU Munich, 81377 Munich, Germany; 7Pettenkofer School of Public Health, 81377 Munich, Germany

**Keywords:** SARS-CoV-2, infections, mental disorders, longitudinal studies, systematic review

## Abstract

In view of disease-related threats, containment measures, and disrupted healthcare, individuals with pre-existing mental illness might be vulnerable to adverse effects of the COVID-19 pandemic. Previous reviews indicated increased mental distress, with limited information on peri-pandemic changes. In this systematic review, we aimed to identify longitudinal research investigating pre- to peri-pandemic and/or peri-pandemic changes of mental health in patients, focusing on the early phase and considering specific diagnoses. PsycINFO, Web of Science, the WHO Global literature on coronavirus disease database, and the Cochrane COVID-19 Study Register weresearched through 31 May 2021. Studies were synthesized using vote counting based on effect direction. We included 40 studies mostly from Western, high-income countries. Findings were heterogeneous, with improving and deteriorating mental health observed compared to pre-pandemic data, partly depending on underlying diagnoses. For peri-pandemic changes, evidence was limited, with some suggestion of recovery of mental distress. Study quality was heterogeneous; only few studies investigated potential moderators (e.g., chronicity of mental illness). Mental health effects on people with pre-existing conditions are heterogeneous within and across diagnoses for pre- to peri-pandemic and peri-pandemic comparisons. To improve mental health services amid future global crises, forthcoming research should understand medium- and long-term effects, controlling for containment measures.

## 1. Introduction

In view of the COVID-19 pandemic as a global public health crisis, patients with pre-existing mental illness have been emphasized as a vulnerable group [1,2,3]. Given the immediate disease-related threats of Severe Acute Respiratory Syndrome Coronavirus 2 (SARS-CoV-2) and pandemic-related measures of containment, this patient group is assumed to be at risk for adverse mental health outcomes [4].

People with pre-existing mental disorders are more vulnerable to SARS-CoV-2 infections, with an increased risk of hospitalization and mortality [5,6,7,8]. Several studies also investigated psychiatric morbidity in the months following a confirmed COVID-19 infection. While Taquet et al. [9] found an increased incidence of psychiatric outcomes (e.g., anxiety or psychotic disorder), Abel et al. [10] identified a similar risk for psychiatric morbidity in cohorts with positive and negative SARS-CoV-2 test results.

Looking at singular mental disorders, one may expect different mental responses to the pandemic, both between individuals with various pre-existing diagnoses as well as within diagnostic groups [11,12]. For example, in view of mandatory social distancing, people with autism spectrum disorder (ASD), who frequently show problems in social interaction, might feel relieved by the reduced exposure to social stressors (e.g., restricted social gatherings). Similarly, people with anxiety disorders, especially those with social anxiety disorder, could feel less burdened. On the other hand, pre-pandemic social anxiety was identified as a predictor of mental symptoms and COVID-19-related worry during the pandemic [13,14]. For people with schizophrenia spectrum and other psychotic disorders, who were shown to have poorer quality social networks [15], mandatory social confinement could result in exacerbated psychotic symptoms and an increase in affective and behavioral symptoms and suicidality [16]. Among patients in recovery from substance use disorder (SUD), social isolation resulting from containment measures is considered a risk factor for relapse [17].

Stay-at-home orders and quarantine during the pandemic could also have a different impact. Given their difficulties adapting to changes, individuals with ASD might be more vulnerable to the development of mental distress [18]. Limited environmental stimuli during stay-at-home orders might elevate the frequency of restricted, repetitive behaviors, potentially interfering with daily functioning. However, based on findings in children with ASD [19], beneficial mental health effects (e.g., being more communicative) are also possible. Given the decreased activity levels, individuals with depressive and bipolar disorders (during depressive episodes) are likely to more often experience feelings of sadness and loss of energy [20]. Similarly, disrupted daily routines might increase the risk of irregular circadian rhythms in individuals with bipolar disorders, with negative consequences for sleep and functioning [21]. In people with eating disorders (anorexia/bulimia nervosa), disruptions of daily routines and restricted physical activities could elevate concerns regarding weight, negatively impacting eating and exercise behavior and potentially increasing eating disorder-specific psychopathology [22]. For people with binge eating disorder, an increased attention toward food (e.g., perceived paucity of food products due to ‘panic buying’ before stay-at-home instructions) might have encouraged food stocking, increasing the likelihood of binge eating [22]. Among individuals with SUD, individuals with opioid disorder receiving methadone treatment, for example, might be unable to access daily clinical treatment during stay-at-home orders and quarantine [17]. Moreover, while external opportunities to gamble declined, people with gambling disorders could be more at risk to gamble online.

Several other pandemic-related stressors could differentially affect mental health in various diagnostic groups. For example, the (over-)exposure to news on the pandemic might intensify (COVID-19-related) fears or even trigger panic episodes in individuals with anxiety disorders [16]. Given the required hygiene behaviors, people with obsessive-compulsive disorder (OCD) could be more likely to experience an elevated level of compulsive thoughts and actions during the pandemic (e.g., handwashing). However, it also seems conceivable that not all compulsive behaviors are affected, and that mental distress partly decreases. For example, OCD patients with fears of contamination could feel more protected than usual since safety behaviors are implemented by many people [23]. Depending on the event initially triggering their traumatic stress reaction, the pandemic and its associated potentially traumatic events (e.g., loss of loved ones, domestic violence during stay-at-home orders) might also increase mental health issues in individuals with pre-existing post-traumatic stress disorder (PTSD), such as intrusion symptoms. To our knowledge, this question has not yet been investigated.

Overall, the unique situation of the COVID-19 pandemic as a global, synchronously starting and persistent macrostressor with multifaceted individual-level stressors raises the questions of whether there has been an initial increase in distress in individuals with pre-existing mental illness, how they have responded over the pandemic course, and whether their mental response has differed depending on the type of diagnosis and within diagnostic groups. To date, the available evidence on psychological effects of the pandemic and associated measures of containment among people with pre-existing mental illness has been synthesized only to a limited extent [4,24,25,26]. Most systematic reviews on mental health amid the pandemic focused on the general population or healthcare workers, with only few studies identified for patients with pre-existing psychiatric disorders [27,28,29,30]. Meta-analyses compared the pre-pandemic situation with peri-pandemic assessments of mental health based on longitudinal research, solely or mostly relying on studies in the general public [31,32] and providing evidence for increased mental symptoms. Several reviews identified the presence of psychiatric illness as a risk factor for mental distress during the pandemic [8,28,33]. Nevertheless, patients with pre-existing mental disorders were studied less frequently so far. In their meta-analysis of cohort studies comparing pre- to peri-pandemic mental health, Robinson et al. [32] found no evidence of changes in symptoms among samples previously diagnosed with mental disorders (25 studies), which is in line with other reviews demonstrating mixed results compared to healthy controls [34]. On the other hand, Neelam et al. [4] and Fleischmann et al. [25] identified more mental symptoms in patients with pre-existing mental illness compared to healthy controls during the COVID-19 and other pandemics, both highlighting the need for longitudinal studies as they could only identify a small number. Similarly, based on 16 longitudinal studies, Carvalho et al. [26] narratively summarized more psychological distress in people suffering from psychiatric disorders compared to healthy controls in the early stages of the pandemic. The authors also reported a condition-specific effect in individuals with eating disorders and obsessive-compulsive disorders. Some reviews specifically focusing on patients with eating, bipolar disorders, or ASD drew similar conclusions [19,35,36,37], although findings were also contradictory. In addition to the lack of longitudinal research, a differentiated perspective on pandemic-related mental health effects depending on the underlying diagnosis of mental illness has mostly not been taken to date [4,25,26].

There is evidence for disruptions in mental healthcare amid the COVID-19 pandemic, with outpatient and community-based mental health services being heavily affected [8,38,39,40,41,42,43]. While outpatient appointments and psychiatric hospitalizations often decreased, the use of mental health emergency consultations was partly found to increase. Furthermore, due to quarantine and travel restrictions, potential barriers of (older) patients with mental disorders in accessing medication and mental healthcare were highlighted [3,44]. These disruptions of mental health services might also have negative long-term consequences for the course and treatment of mental illness beyond the pandemic.

Based on the potential psychological impact of the COVID-19 pandemic among people with pre-existing mental disorders and the presented evidence gaps, a systematic synthesis of the current literature seemed needed. The objective of the current review was therefore to extend insights gained from previous research by identifying longitudinal and repeated cross-sectional observational studies measuring mental health in this group. Given the difficulties of people with many mental disorders to adapt to new situations, we specifically focused on the period shortly after the pandemic outbreak and the further pandemic course. We aimed at investigating pandemic-related mental health effects depending on different diagnoses by summarizing the evidence for pre- to peri-pandemic as well as peri-pandemic mental health changes.

## 2. Materials and Methods

Search strategy. This systematic review adheres to standards of conduct and reporting as outlined in the Cochrane Handbook [45] and the Preferred Reporting Items for Systematic Reviews and Meta-Analyses (PRISMA) [46]. Differences between the preregistered protocol (PROSPERO CRD42021251770) and the review are presented in the Appendix A. Four electronic databases were searched from inception to 31 May 2021: PsycINFO Ovid, Web of Science (Science Citation Index and Emerging Citation Index), WHO Global literature on coronavirus disease database, and the Cochrane COVID-19 Study Register (comprising the Cochrane Central Register of Controlled Trials, PubMed, Embase, ClinicalTrials.gov, WHO International Clinical Trials Registry Platform, medRxiv). The search was developed by an experienced information specialist (MIM) with quality assessment by a second information specialist (for details of search strategy see Appendix A). No restrictions were placed on publication date or language.

Selection criteria. Eligible studies were longitudinal or repeated cross-sectional observational studies, which could have either one or more pre-pandemic and one or more peri-pandemic assessments or multiple assessments during the COVID-19 pandemic. Included studies provided quantitative data on disorder-specific (e.g., depressive symptoms) and/or general mental health outcomes (e.g., psychological distress) among adults 18 years and older with (pre-existing) mental disorders (see Appendix A). Study participants had to be diagnosed with mental illness according to ICD-10 (International Classification of Diseases) or DSM-5 (Diagnostic and Statistical Manual of Mental Disorders) classification systems before 11 March 2020 (i.e., World Health Organization declaration of pandemic). Inpatient and outpatient study settings and different states of mental illness (e.g., severity, incipient/acute vs. chronic vs. remission, comorbidities) were eligible. Publications had to describe that (i) participants were diagnosed based on a structured clinical interview (e.g., Mini-International Neuropsychiatric Interview (M.I.N.I.)) or that (ii) they had begun psychiatric and/or psychotherapeutic treatment before the pandemic (i.e., including pretreatment diagnostics). We only included published studies (i.e., no study protocols or preprints).

Study selection. Following deduplication, titles/abstracts were screened by two reviewers, respectively, who worked independently (screening team: AMK, CK, MC, NR, RME, and SL) using EPPI-Reviewer [47], with two mechanisms of machine learning (see Appendix A). Inter-rater reliability (kappa = 0.53) was fair. At the full-text level, eligibility of remaining articles was double-screened by two reviewers, respectively (screening team: CK, MC, NR, RME, and SL), resulting in good inter-rater reliability (kappa = 0.61). At both stages of screening, disagreements were resolved through discussion or by consulting a third reviewer.

Data extraction. A customized data extraction sheet was developed (see Appendix A). Data were extracted by one reviewer (CK, MC, NR, RME, or SL) in Excel and checked by a second reviewer. Any disagreements were resolved by discussion or by consulting a third reviewer.

Quality assessment. The quality of included studies was assessed by adapting the National Institute of Health (NIH) Quality Assessment Tool for Observational Cohort and Cross-Sectional Studies [48]. Four items of the original tool were removed due to lack of fit with pandemic-related mental health research. Two items were adapted, while eight items were not modified. Overall, ten items were applied (see Appendix A). Items were primarily answered with ‘yes’ or ‘no’, with subsequently added options (e.g., ‘not reported’) for some items. Following the recommendation for the original NIH tool, we deemed an overall judgement of study quality as inappropriate. Quality assessment was performed by one reviewer (NR) and checked by a second reviewer (MC, SL), with disagreements being resolved through discussion.

Data synthesis. Pairwise meta-analyses for various mental health outcomes according to different diagnoses were planned (see PROSPERO CRD42021251770). However, due to methodological heterogeneity of eligible studies in terms of analytical methods, outcomes, and reporting (see Appendix A), we deemed a meta-analytic approach to be inappropriate. Instead, we conducted a synthesis without meta-analysis (SWiM) following the respective guidance [49] and the recommendations of the Cochrane Handbook [50]. To perform narrative synthesis, included studies were clustered by the reported comparison of mental health (i.e., pre- to peri-pandemic or peri-pandemic changes), the underlying diagnosis of mental illness according to the DSM-5 classification system, and mental health outcome. We used vote counting based on the effect direction for different outcomes reported in each included study; this approach, and specifically our classification for direction of effect, is described below in Table 1. Based on the available evidence and DSM-5, we created nine diagnostic groups (i.e., ASD, schizophrenia and other psychotic disorders, bipolar disorders, depressive disorders, anxiety disorders, OCD, PTSD, eating disorders, and substance-related and addictive (gambling) disorders) and one mixed group that were used for vote counting. For example, the group of anxiety disorders comprised individuals with various diagnoses of anxiety disorder, including social anxiety disorder, panic disorder, agoraphobia, and generalized anxiety disorder (GAD).

Due to their importance for several mental disorders (i.e., transdiagnostic relevance) and since we considered them as indicators of overall mental health, general outcomes (e.g., anxiety, depressive symptoms, psychological distress, stress) were synthesized independent of the number of available studies. For disorder-specific outcomes (e.g., eating disorder-specific psychopathology), vote counting was limited to outcomes assessed by at least two studies within a diagnostic group and across both types of comparisons (i.e., pre- to peri-pandemic and peri-pandemic; see Appendix A). If studies provided data on the same outcome in multiple groups of patients with mental illness (e.g., panic disorder and social phobia within anxiety disorders), all available comparisons were considered for narrative synthesis. The same applied to studies reporting multiple time comparisons of an outcome within the category of pre- to peri-pandemic or peri-pandemic changes of mental health (e.g., difference between two pre-pandemic and one peri-pandemic assessment, respectively).

## 3. Results

### 3.1. Study Selection and Characteristics of Included Studies

The search yielded a total of 14,753 records. After deduplication, 10,097 records were screened at the title/abstract stage. Of these, 9410 were deemed not relevant based on title/abstract, meaning that 687 full-text articles were assessed. At the full-text level, 643 ineligible reports were excluded. For this review, 40 studies (from 41 reports) were eligible for inclusion (see Figure 1).

Characteristics of included studies are presented in Table 2 (details in Appendix A). Most studies used longitudinal designs measuring pre- to peri-pandemic changes in mental health (*k* = 17), peri-pandemic changes (*k* = 16), or both (*k* = 4), while three were repeated cross-sectional studies (pre- to peri-pandemic: *k* = 2; peri-pandemic: *k* = 1). The majority of studies was conducted up to June 2020, with only a limited number of studies assessing later phases [51,52,53,54,55,56,57,58], and Peckham et al. [59] covered the longest survey period (until December 2020). Most included studies were conducted in European countries (*k* = 19) and the United States (*k* = 11), with only few available studies from Asian countries (*k* = 8), a single study from Australia, and one international study. Studies were primarily performed in outpatient (*k* = 14) or mixed settings (i.e., inpatient and outpatient; *k* = 5) and in patients who had received previous treatment (e.g., psycho-/pharmacotherapy; *k* = 31). Participants were predominantly female (range: 9.1% to 100%) and young-to-middle-aged adults (*k* = 31; mean age range: 22.7 to 55.7 years), with only few studies investigating older adults ≥ 60 years; [55,60,61,62,63], and sample sizes ranged from 11 to 2013 participants. Regarding the chronicity of mental illness, most participants had been diagnosed before the age of 18 years, total illness duration was longer than two years, they were in treatment for more than one year, or they lived in residential care.

Of the 40 studies, 11 examined patients with various diagnoses of mental illness, while 29 focused on specific disorders. If possible, data from studies reporting on several diagnostic groups separately were assigned to the respective diagnostic group [58,64,65]. Eight studies reported mental health data in individuals with schizophrenia spectrum and other psychotic disorders. Seven included patients with OCD, followed by six studies in people with eating disorders, five among individuals with ASD, and four studies in people diagnosed with bipolar disorders. Patients with SUD and depressive disorders were investigated in three studies, respectively. Data on patients with anxiety disorders and PTSD were reported by two studies apiece, while a single study examined individuals with gambling disorders. Data presented in eight studies could not be assigned to specific diagnoses, resulting in a mixed category [52,54,55,59,63,65,66,67]. All studies used self-reported mental health outcomes, using screening tools to assess general and disorder-specific symptoms.

**Table 2 ijerph-20-00948-t002:** Study characteristics of included studies (summary).

Study ID	Diagnostic Groups	Country	Study Design	Setting Category	Diagnosis	Previous Treatment/Duration of Illness	Sample Size/n(%) Female/Age (*M* ± *SD*)	Assessments	Outcomes
**A. Pre- to peri-pandemic changes of mental health**									
Castellini 2020 [68]	ED	Italy	L	Outpatient	C	Yes/NR	74/74 (100%)/31.74 ± 12.76	P1: January 2019–September 2019 P2: November 2019–January 2020 D1: 22 April 2020–3 May 2020	Eating disorder specific psychopathology; general psychopathology; objective binge eating monthly
Chakraborty 2020 [23]	OCD	India	L	Outpatient	NR; probably C	Yes/NR	84/64 (76.2%)/NR	P1: NR (pre-pandemic scores; last recorded Y-BOCS severity score noted from the case register) D1: 23 April 2020–22 May 2020	OCD severity
Cordellieri 2021 [69]	SSOPD	Italy	L	Residential care	NR; probably C	Yes/NR (chronic, residential living)	22/10 (45.5%)/31.82 ± 6.96	P1: November 2019 D1: April 2020	Psychiatric symptoms
Giel 2021 [51]	ED	Germany	L	NR	C	Yes/duration of illness (from original IMPULS trial 3 years before): IG: 15.9 ± 11.4, CG: 15.5 ± 12.2	42 (52% of IMPULS trial sample)/34 (81%)/41.1 ± 12.6 (at baseline), 41.3 ± 12.6 (end of treatment), 45.5 ± 12.6 (COVID-19 follow-up)	P1: NR (entering IMPULS trial; IMPULS trial carried out between March 2015–September 2017) P2: NR (end of treatment/trial) D1: May 2020–July 2020	Depressive symptoms; eating disorder specific psychopathology; general psychopathology
Goldfarb 2022 [70]	ASD	Israel	L	Other	C	NR/NR	23 (completed both surveys)/4 (17.4%)/30.22 ± 7.4	P1: September 2019–January 2020 D1: April 2020–May 2020	Psychological (emotional) distress
Hamm 2020 [60]	DD	USA	L	Probably outpatient	C	Yes/38.6 years (age at survey M ± SD = 69.2 ± 6.0; age at onset 30.6 ± 18.8)	73/50 (68.5%)/ 69.2 ± 6.0	P1: ~April 2019 ± 9 months (anxiety symptoms), ~May 2019 ± 8 months (depressive symptoms) P2: ~December 2019 ± 5 months (depressive symptoms) D1: 1 April 2020–23 April 2020	Anxiety symptoms; depressive symptoms
Johnco 2021 [63]	Mixed (anxiety and/or depressive disorder); no subgroup data reported	Australia	L	NR	C	Yes/on average 5.67 years post-treatment	37/24 (65%)/75 ± 5	P1: ~2009–2019 (1 to 129 (M = 68, SD = 43) months prior to COVID-19 lockdown) D1: April 2020–May 2020	Anxiety symptoms; depressive symptoms; psychological distress; quality of life (psychological health, social relationships)
Khosravani 2021 [53]	OCD	Iran	L	NR	C	Yes/age of OCD onset: 26.6 ± 8.55; illness duration: 9.6 ± 7.0 years	270/155 (57.4%)/36 ± 12.1	P1: NR (before outbreak of COVID-19) D1: May 2020–July 2020	OCD severity
Kott 2020 [71]	SSOPD	International/NR	R-CS	NR	NR	NR/NR	NR/NR/NR	P1: NR (before the date of the first confirmed COVID-19 case within each country) D1: NR (after the date of the first confirmed COVID-19 case within each country)	Anxiety, excitement, hallucinatory behavior
Machado 2020 [72]	ED	Portugal	L	Probably clinical-acute	NR; probably C	Yes/NR	43/41 (95.3%)/27.60 ± 8.45	P1: NR (last available evaluation before COVID-19 lockdown period) ^1^ D1: 30 April 2020–15 May 2020	Eating disorder psychopathology
Matsunaga 2020 [73]	OCD	Japan	L	NR	C	Yes/>3 years	60 (total), of these: fully remitted persons: *n* = 24,partially remitted persons: *n* = 36/total sample: 35 (58.3%)/total sample: 41.5 ± 7.9	P1: NR (before December 2019) D1: 7 April 2020–2 May 2020	OCD severity
Orhan 2021 [62]	BD	Netherlands	L	Probably outpatient	C	NR/NR (chronic: >2 years; see recruitment)	81/45 (55.6%)/66.1 ± 7.2	P1: 2017–2018 D1: April 2020	Anxiety symptoms; depressive symptoms; (hypo-)maniac symptoms; loneliness
Pan 2021 [64]	OCD; DD; AD (mixed but also subgroup data reported for three different diagnoses and subdiagnoses along with total patient sample)	Netherlands	L	NR	C	Yes/NR	Overall: 1517, lifetime mental health disorder: 1181 (77.9%), panic disorder: *n* = 428–481; generalized anxiety disorder: n = 413–456; agoraphobia: *n* = 360–404; social anxiety disorder: *n* = 465–523; major depressive disorder: *n* = 852–984; dysthymic disorder: *n* = 336–373; obsessive compulsive disorder: *n* = 120–124 (only outcomes anxiety and loneliness assessed in this subgroup). Overall: 976 (64%); patients: 791 (67%); Overall: 56.1 ± 13.2; patients: 55.7 ± 12.9 ^2^	P1: 2006–2016 D1: 1 April 2020–13 May 2020	Anxiety symptoms; depressive symptoms; loneliness
Peckham 2021 [59]	Mixed (schizophrenia or delusional/psychotic illness or bipolar disorder), but no subgroup data reported	UK	L	NR	C	NR/NR	367/180 (49%)/50.5 (SD NR)	P1: April 2016–March 2020 D1: July 2020–December 2020	/
Pinkham 2020 [65]	SSOPD (mixed for ‘affective disorder’, i.e., bipolar and depressive disorders, but also subgroup data for schizophrenia spectrum reported)	USA	L	Outpatient	C	NR/NR	Schizophrenia spectrum: 92; affective disorders: 56/schizophrenia spectrum: 50 (54.3%); affective disorders: 40 (71.4%)/schizophrenia spectrum: 42.95 ± 10.76; affective disorders: 40.77 ± 11.76	P1: 4 December 2018–4 January 2019 (study 1) and 11 July 2019–21 July 2019 (study 2); pre-pandemic symptom severity averaged across all completed surveys D1: 3 April 2020–4 June 2020	Energized/excited; hearing voices; sad/depressed; sleep (in hours); substances used; well-being
Rutherford 2021 [61]	PTSD	USA	L	Probably outpatient	C	NR/chronic PTSD (duration at least 6 months)	46/25 (54.3%)/62.5 ± 9.0	P1: before 13 March 2020 D1: 1 April 2020–8 May 2020	Depressive symptoms; post-traumatic stress symptoms
Seitz 2021 [67]	Mixed (PTSD, MDD, SSD; no subgroup data reported)	Germany	L	Mixed inpatient/outpatient	C	Yes/NR	63/NR: Only provided for full sample (individuals with psychiatric disorders and healthy volunteers)/NR	P1: September 2018–November 2019 D1: 16 April 2020–18 May 2020	General psychopathology
Sharma 2021 [74]	OCD	India	R-CS	Outpatient	C	Yes/age at onset of OCD: pandemic cohort: 21.44 ± 8.52; historical cohort: 21.83 ± 8.53; duration of illness: pandemic cohort: 10.92 ± 7.41, historical cohort: 11.14 ± 7.98	Pandemic cohort: 240/89 (37%)/32.28 ± 9.70; historical cohort (data from medical records of independent set of OCD patients followed up during same period 1 year prior): 207/72 (34.8%)/32.97 ± 11.14	Historical control cohort: P1: NR, baseline assessment (first visit to OCD clinic) P2: 1 October 2018–28 February 2019 (FU visit in clinic; 1 year prior to FU visit of pandemic cohort) P3: April 2019–May 2019 (2nd FU visit in clinic; 1 year prior to 2nd FU visit of pandemic cohort) Pandemic cohort: P1: NR, baseline assessment (first visit to OCD clinic) P2: 1 October 2019–29 February 2020 ^3^ (FU visit in clinic before pandemic) D1: 26 April 2020–12 May 2020 (telephonic FU during pandemic)	OCD severity
Strauss 2022 [57]	SSOPD	USA	L	Outpatient	C	Yes/NR (chronic)	32 patients with chronic schizophrenia or schizoaffective disorders/patients: 24 (75%)/patients: 40.13 ± 13.25	P1: ~August–November 2018 ± 6 monthsD1: July 2020–October 2020	Motivation and pleasure
**B. Peri-pandemic changes of mental health**									
Adams 2021 [75]	ASD	USA	L	NR	C	Yes/age of autism diagnosis/at onset: 8.71 ± 4.66; duration: 17.74	322 (participants consented); 315 completed survey at D1; 275 completed survey at D2; 275 completed both surveys + were analyzed/135 (49.1%)/26.45 ± 4.66	D1: 11 March 2020–20 March 2020 D2:18 May 2020–27 May 2020	Anxiety symptoms; depressive symptoms; stress
Bal 2021 [76]	ASD	USA	L	NR	Self-reported	Yes/pre-pandemic: 83.8% with previous mental health diagnosis, 42% with diagnosis before 18 years	396 (completed both surveys)/233 (58.8%)/37.38 ± 13.36	D1: 30 March 2020–10 April 2020 D2: 27 May 2020–6 June 2020	Psychological distress
Brondino 2020 [77]	ASD	Italy	L	Daycare	C	Yes/NR	18/5 (27.8%)/22.72 ± 4.75	D1: 19 February 2020 ^4^ D2: 4 March 2020	Psychiatric symptoms and problem behaviors
Carta 2021 [78]	BD	Italy, Tunisia	L	Probably outpatient	C	Yes/≥1 year (in treatment)	Cagliari (exposed to rigid lockdown): 40/28 (70%)/48.57 ± 11.64; Tunis (less severe lockdown/not exposed): 30/16 (53.3%)/41.8 ± 13.22	D1: April 2020 D2: June 2020	Circadian rhythms; depressive symptoms
Daly 2021 [58]	SSOPD; OCD; ED; BD; DD; AD; PTSD (mixed but also subgroup data reported)	USA	L	NR	Self-reported	NR/NR	Of 7319 patients in total sample, 27.5% (n = 2013) reporting pre-existing mental health diagnoses; anxiety disorders: 16.2% (*n* = NR); bipolar disorders: 3.4% (*n* = NR); depressive disorders: 19.1% (*n* = NR); eating disorders: 1.9% (*n* = NR); obsessive compulsive disorders: 3.1% (*n* = NR); post-traumatic stress disorders: 6.4% (*n* = NR); schizophrenia/psychotic disorders: 0.8% (*n* = NR)/3755 (51.3%)/48.9 ± 16.5 ^2^	D1: 10–18 March 2020 D2: 1–14 April 2020 D3: 15–28 April 2020 D4: 29 April–12 May 2020 D5: 13–26 May 2020 D6: 27 May–9 June 2020 D7:10–23 June 2020 D8: 24 June–20 July 2020	Psychological distress
Davide 2020 [79]	OCD	Italy	L	Outpatient	C	Yes/NR	30/16 (53.33%)/43.17 ± 14.87	D1: January 2020–February 2020 ^4^ D2: 16–17 April 2020	OCD severity
Donati 2021 [80]	GD	Italy	L	Outpatient	C	Yes/7 years on average (under treatment)	135/26 (19%)/50.07 ± 13.33	D1: NR (before lockdown) ^4^ D2: 7 April 2020–28 May 2020	Gambling problem symptoms (total score)
Gaume 2021 [81]	SUD	Switzerland	L	Mixed inpatient/outpatient	NR; probably C	Yes/NR	D1: 49; D2: 51/D1: 11 (22.9%); D2: 12 (27.3%)/D1: *Mdn* (*IQR*): 39 (32–50); D2: 41 (34–48)	D1: 17–24 April 2020 D2: 4–8 May 2020	Use of heroin/cocaine/cannabis; impact of pandemic on alcohol use/use of prescription drugs/other drugs; impact of pandemic on mental health in general
Hennigan 2021 [52]	Mixed; various anxiety disorders and OCD; no subgroup data reported)	Ireland	L	NR	C	Yes/NR	Total (mixed): 24/16 (66.7%)/37.4 ± 11.4	D1: ~April 2020 (appr. 6 months before assessment 1) ^5^ D2: 15 October 2020–29 October 2020	Anxiety symptoms; global functioning; impact of COVID-19 pandemic on mood symptoms, social functioning, quality of life.
Hochstatter 2021 [82]	SUD	USA	R-CS	Other, mobile health application	NR; probably C	Yes/NR	64/16 (25%)/49 (NR)	D1: 31 January 2020–12 March 2020 ^6^ D2: 24 March 2020–4 May 2020	Alcohol use/marijuana use/other illicit drugs
Leenaerts 2021 [83]	ED	Belgium	L	NR	C	Yes/maximum illness duration of 5 years (inclusion criterion); *Mdn* = 3 (*IQR*: 2–5) years	15/15 (100%)/*Mdn* (*IQR*) = 23 (4)	D1: 10 January 2020–14 March 2020 D2: 19 March 2020–9 May 2020	Loss of control over their eating behavior/binge eating episode
Lugo-Marín 2021 [18]	ASD	Spain	L	NR	C	Yes/NR (chronic; see recruitment)	35/12 (34.3%)/32.8 ± 13.1	D1: NR (pre-lockdown) ^7^ D2: NR (postlockdown; 8 weeks after lockdown onset)	Anxiety symptoms; depressive symptoms; psychological distress
Ma 2020 [84]	SSOPD	China	L	Clinical-chronic/Rehabilitation	C	Yes/course of psychosis (years): isolation group: 20.20 ± 9.26 years; long-term hospitalization (length of hospitalization): isolation group: 4.90 ± 2.67 years	30/18 (60%)/43.17 ± 11.55	D1: NR (before isolation) D2: assessments on 10th–14th days of isolation (isolation period for 30 participants: 10 January 2020–30 April 2020)	Anxiety symptoms; depressive symptoms; psychological stress; severity of participants’ psychiatric symptoms; sleep quality
Ma 2021 [85]	SSOPD	China	L	Clinical-chronic/Rehabilitation	C	Yes/course of schizophrenia: M ± SD = 6.8 ± 5.6 years	21/12 (57.1%)/43.1 ± 2.6	D1: 01/2020 (patients were uninfected) ^8^ D2: NR (within 3 days of diagnosis with COVID-19 after patient was transferred to isolation ward) D3: NR (after patients were cured; before they were transferred out of isolation ward; transfer of last cured patient on 30 March 2020)	Psychopathology; stress
Nisticò 2021 [86]	ED	Italy	L	Outpatient	C	Yes/NR	59/57 (97%)/30.1 ± 12.9	D1: 25 April 2020–28 April 2020 D2: 25 June 2020–28 June 2020	Anxiety symptoms; depressive symptoms; psychological distress; losing control over food; stress; well-being
Seethaler 2021 [55]	Mixed (affective or anxiety disorders; no subgroup data reported)	Germany	L	Mixed inpatient/outpatient	C	Yes/NR	D1: 32; D2: 24/D1: 20 (62.5%); D2: 16 (66.67%)/D1: 77.94 ± 8.12; D2: 78.25 ± 8.43	D1: April 2020–May 2020 D2: August 2020	Anxiety; current suicidality (suicidal thoughts, suicidal plans, suicidal attempt); depressive symptoms; severity of illness
Wynn 2021 [56]	SSOPD	USA	L	NR	C	NR/NR	81/9 (11.1)/54.4 ± 9.8	D1: mid May 2020–mid August 2020 D2: mid August 2020–mid October 2020	Alcohol use/cannabis use; anxiety symptoms; depressive symptoms; loneliness; motivation and pleasure
**C. Both pre- to peri-pandemic and peri-pandemic changes of mental health**									
Liu 2021 [87]	SUD	China	L	Outpatient	C	Yes/course of drug addiction: 22.32 ± 8.42 years	76/26 (34.2%)/48.53 ± 5.99	P1: October 2019–December 2019 D1: February 2020–April 2020 (outbreak) D2: May 2020–June 2020 (postpandemic)	Anxiety symptoms; alcohol consumption; amphetamine/morphine use; depressive symptoms; stress; tobacco consumption
Mergel 2021 (follow-up study); Schützwohl 2020 (original study reporting two assessments) [54]	Mixed (schizophrenia, affective disorders, anxiety disorders, personality disorders; no subgroup data reported but separate data for patients with chronic and acute mental disorders)	Germany	L	Group 1 (chronic mental disorders): Residential care Group 2 (acute mental disorders): Clinical-acute	C	Yes/chronicity: group 1 (chronic): 6–10 years: 26.1%, >10 years: 73.9%; acute group (group 2): <2 years: 4%, 2–5 years: 16%, 6–10 years: 24%, >10 years: 56%	Total: P1: 174; D1: 132; D2: 106; group 1 (chronic mental disorders; not acutely ill in 4 weeks prior to initial survey): n = 19–27; group 2 (acute mental disorders): n = 26–30/group 1: 13 (48.1%); group 2: 18 (60.0%)/group 1: 49.7 ± 13.1; group 2: 44.0 ± 11.8	P1: August 2019–March 2020 D1: 23 March 2020–20 April 2020 D2: 22 June 2020–19 July 2020; first two assessments reported in Schützwohl 2020	Anxiety symptoms; depressive symptoms; perceived impairments in social participation (close personal relationships, stress and extraordinary strain); psychological distress
Riblet 2021 [66]	Mixed (MDD; BAD type I, Psychotic Disorder, PTSD, PD, agoraphobia, GAD, SAD, OCD, AUD, SUD), but no subgroup data reported	USA	L	Outpatient	C	Yes/NR	11/1 (9.1%)/48.0 ± 17.7	P1: October 2019–December 2019 P2: November 2019–January 2020 D1: February 2020–March 2020 ^9^ D2: 23 April 2020–04 May 2020	Hopelessness; suicidal ideation; thwarted belongingness
Yocum 2021 [88]	BD (mixed; mostly bipolar disorders)	USA	L	Mixed inpatient/outpatient	C	NR/NR	560 (total), 345 (participants with bipolar disorder), 147 (healthy controls)/381 (68%) of total sample (bipolar + healthy controls), NR for bipolar patients/49 (NR) of total sample (bipolar + healthy control), NR for bipolar patients	For pre- to peri-pandemic changes: P1: 15 March 2015–2019 to 30 May 2015–2019 D1: 15 March 2020–30 May 2020 For peri-pandemic changes: D1: 30 April 2020 D2: 14 May 2020 D3: 28 May 2020	Anxiety symptoms (pre- vs. during and peri-pandemic); degree of change/disruption due to COVID-19 pandemic (experiencing pandemic-related stress; only peri-pandemic); depressive symptoms (pre- vs. during and peri-pandemic); sleep (% bad sleep quality; pre- vs. during and peri-pandemic)

Note. *Abbreviations:* ~: approximately; AD: anxiety disorder; ASD: autism spectrum disorder; AUD: alcohol use disorder; BAD: bipolar affective disorder; BD: bipolar disorder; C: clinician-based (i.e., ICD-10/DSM-5); D: during COVID-19 assessment (e.g., D1: first during-COVID-19 assessment); DD: depressive disorder; DSM-5: Diagnostic and Statistical Manual of Mental Disorders-5; ED: eating disorder; GAD: generalized anxiety disorder; GAD-7: general anxiety disorder; GD: gambling disorder; ICD-10: International Classification of Diseases; *IQR*: interquartile range; L: longitudinal; *M*: mean; *Mdn*: median; MDD: major depressive disorder; NR: not reported; OCD: obsessive-compulsive disorder; P: pre-COVID-19 assessment (e.g., P1: first pre-COVID-19 assessment); PD: panic disorder; PTSD: post-traumatic stress disorder; R-CS: repeated cross-sectional; SAD: social anxiety disorder; SCID: Structured Clinical Interview for DSM-5; *SD*: standard deviation; SSD: somatic symptom disorder; SSOPD: schizophrenia spectrum and other psychotic disorder; SUD: substance use disorder; UK: United Kingdom; USA: United States of America. ^1^ First COVID-19 case on 2 March 2020; first lockdown in Portugal on 18 March 2020. ^2^ Proportion of female participants and age NR for various subgroups with different diagnoses of mental disorders. ^3^ First COVID-19 case in India on 30 January 2020. ^4^ First COVID-19 case in Italy on 29 January 2020; first ‘lockdown’ in Italy from 9 March 2020 onwards. ^5^ First COVID-19 case in Ireland on 1 March 2020. ^6^ See publication: First COVID-19 case in Wisconsin on 5 February 2020; first COVID-19 case in the USA on 20 January 2020. ^7^ First COVID-19 case in Spain on January 20, 2020; First lockdown in Spain declared on 14 March 2020. ^8^ First COVID-19 case in China on 4 January 2020; isolation wards officially established on 30 January 2020. ^9^ First COVID-19 case in the USA on 20 January 2020.

### 3.2. Quality Assessment

Appendix A presents the quality assessment based on the modified NIH tool. Research questions, study populations, eligibility criteria, pandemic exposure, and outcome measures were clearly described by all or most of the included studies, with good psychometric quality for most measures. Study quality was heterogeneous regarding the consistent exposure to the COVID-19 pandemic among all participants (i.e., assessments over period of more than four weeks with probably varying containment measures). We found mixed quality concerning loss to follow-up. Most limitations in terms of quality were found for the domains of a priori sample size calculation and the resulting power, which were hardly reported, the repeated assessment of exposure (i.e., mostly maximum of two peri-pandemic assessments), and the adjustment for potential confounding variables.

### 3.3. Mental Health Effects of the COVID-19 Pandemic Compared to Pre-Pandemic Baseline and Peri-Pandemic Changes of Mental Health

Effects from included studies for each outcome in each diagnostic group, based on the effect direction in available mental health comparisons, that is, pre- to peri-pandemic or peri-pandemic changes, are presented in Table 3. The underlying statistical data informing these effect directions are summarized in Appendix A. Figure 2 provides a graphical summary of the results across all diagnostic groups.

#### 3.3.1. Autism Spectrum Disorders

While mental health data for pre- to peri-pandemic comparisons were poorly represented for individuals with ASD (Table 3), effect directions for peri-pandemic changes consistently indicated a (potential) improvement of mental health. Goldfarb et al. [70] reported a single effect suggesting lower pre-pandemic psychological distress (↓). Effect directions for peri-pandemic changes were classified as (un-)clear effects, (potentially) indicating an improvement of anxiety (↗: in two of two available comparisons, i.e., 2/2), depressive symptoms (↑: 1/2; ↗: 1/2), psychological distress (↑: 1/3; ↗: 2/3), and stress (↗: 1/2) in the pandemic course.

#### 3.3.2. Schizophrenia Spectrum and Other Psychotic Disorders

A large variety of effect directions was observed, and findings were mixed for both pre- to peri-pandemic and peri-pandemic comparisons of mental health (Table 3). Clear and unclear effect directions (potentially) indicated better pre-pandemic values of anxiety (↓: 1/2) and depressive symptoms (↘: 1/1), diminished motivation and pleasure (↓: 1/1), and diminished sleep quantity (↘: 1/1). For well-being, Pinkham et al. [65] reported a clear effect (↑: 1/1) indicating an improvement of well-being. The body of evidence was mixed for the outcomes of excitement/energization (↑: 2/3; ↓: 1/3), hallucinations (↑: 1/3; ↗: 1/3; ↓: 1/3), and substance use (↑: 1/2; ↘: 1/2). Regarding peri-pandemic changes, two studies assessed mental health in schizophrenia patients subjected to social isolation after contact with COVID-19 patients [84] and hospitalized schizophrenia patients with COVID-19 [85]. After isolation, Ma et al. [84] observed (un-)clear effects, (potentially) indicating increased levels of anxiety (↓), depressive symptoms (↘), stress (↓), and decreased sleep quality (↓), while there was one unclear effect potentially suggesting reduced psychological distress during isolation (↗). Ma et al. [85] provided mixed effect directions, with psychological distress potentially decreasing and stress increasing between patients’ noninfection and infection/isolation, followed by a possible increase in distress and a reduction of stress in patients moving from being infected/isolated to being cured from COVID-19. Remaining (clear/unclear) effect directions for peri-pandemic changes, mainly reported by Wynn et al. [56], (potentially) suggested an improvement during the pandemic for anxiety and depressive symptoms (↑), loneliness (↑), psychological distress (↗), and substance use (↗). For diminished motivation and pleasure, there was one effect potentially indicating a deterioration of mental health (↘).

#### 3.3.3. Bipolar Disorders

In patients with a pre-existing diagnosis of bipolar disorder, both findings for pre- to peri-pandemic and peri-pandemic changes were heterogeneous (Table 3). Observed effects for pre- to peri-pandemic comparisons indicated decreased peri-pandemic levels of mental symptoms (↑; anxiety: 1/2, depressive symptoms: 1/2, (hypo-)maniac symptoms: 1/1), potentially improved peri-pandemic sleep quality (↗: 1/1), and a null effect (∎; loneliness: 1/1), but also potentially better pre-pandemic mental health (↘; anxiety: 1/2; depressive symptoms: 1/2). Around 62% of available effects for peri-pandemic changes were unclear effects, potentially suggesting an improvement of mental health in the pandemic course (↗). The proportion of this effect direction was 2/2 for anxiety, 3/4 for depressive symptoms, 2/2 for sleep quality, and 1/2 for stress. Remaining effects potentially indicated poorer mental health in the pandemic timeline (↘) for circadian rhythms (1/1), psychological distress (2/2), and, partly, stress (1/2).

#### 3.3.4. Depressive Disorders

While the majority of effects for pre- to peri-pandemic changes in individuals with depressive disorder indicated better peri-pandemic mental health, the evidence on peri-pandemic changes was limited (Table 3). Based on two studies [60,64], most observed effects on anxiety (↑: 1/3; ↗: 1/3) and depressive symptoms (↑: 2/4; ↗: 1/4) were classified as (un-)clear effects, (potentially) suggesting a pre- to peri-pandemic reduction of symptoms. Effects in these categories were consistently found for people with persistent depressive disorder (i.e., dysthymia [64]). Only two unclear effects, both in patients with major depressive disorder (MDD [64]), potentially suggested better pre-pandemic values of anxiety and depressive symptoms. Findings for loneliness were heterogeneous. For peri-pandemic changes, Daly and Robinson [58] was only available study that reported an increase in distress in individuals with depression from March–April 2020 (↓), followed by a recovery further along in the timeline (↑; April–July 2020).

#### 3.3.5. Anxiety Disorders

Effect directions indicated potentially better peri-pandemic mental health compared to pre-pandemic data at least for some diagnoses of anxiety disorder, while evidence on peri-pandemic changes was poorly represented (Table 3). Specifically, most effects in Pan et al. [64] were unclear (↗), potentially indicating lower peri-pandemic levels of anxiety (3/4), depressive symptoms (4/4), and loneliness (3/4). While these effects were consistently observed for individuals with GAD and social anxiety disorder, results were slightly more heterogeneous for people with panic disorder (↗: 2/3; ↘: 1/3 for loneliness) and agoraphobia (↗: 2/3; ↘: 1/3 for anxiety symptoms). For peri-pandemic changes, Daly and Robinson [58] was only available study that reported an increase in distress from March–April 2020 (↓), followed by a recovery in the further timeline (↑; April–July 2020).

#### 3.3.6. Obsessive-Compulsive Disorders

Both the evidence for pre- to peri-pandemic and peri-pandemic changes of mental health was limited for patients with OCD (Table 3). While Pan et al. [64] found one effect potentially indicating higher peri-pandemic values of anxiety (↘), another effect in this study potentially suggested a lower peri-pandemic level of loneliness (↗). Regarding the severity of obsessive-compulsive symptoms (OCD severity), vote counting was mostly limited due to the lack of statistical testing for differences (see Appendix A), although there was also one effect indicating increased peri-pandemic OCD severity (↓). For peri-pandemic changes, Daly and Robinson [58] identified a potential increase in psychological distress (↘) from March–April 2020, which was followed by a recovery further along in the pandemic timeline (↑; April–July 2020). The single clear effect (↓) found by Davide et al. [79] also indicated an increase in OCD severity between January/February and April 2020, although the first assessment was very close (and possibly even before for some participants) to the first COVID-19 case in Italy (29 January 2020).

#### 3.3.7. Post-Traumatic Stress Disorders

Overall, the evidence base for patients with pre-existing PTSD was very limited (Table 3), with findings being mixed for pre- to peri-pandemic comparisons but in line with other diagnostic groups for peri-pandemic changes, at least based on a single study. Single effects each indicated both a decrease in post-traumatic stress symptoms (↑: 1/1) and a potential increase in depressive symptoms (↘: 1/1) compared to pre-pandemic data. Daly and Robinson [58] found effects potentially suggesting a deterioration of psychological distress during an early phase of the pandemic (March–April 2020), followed by a potential improvement in the further course until summer 2020.

#### 3.3.8. Eating Disorders

For individuals with pre-existing eating disorders, a variety of effect directions was observed, with most of them indicating better peri- than pre-pandemic mental health and a possible improvement of mental health over the pandemic course (Table 3). Most effect directions for pre- to peri-pandemic comparisons were categorized as clear or unclear effects, (potentially) suggesting reduced levels of eating disorder-specific psychopathology (↑: 3/7; ↗: 1/7), loss of control overeating behavior (↑: 1/4; ↗: 1/4), and psychological distress (↑: 3/6; ↗: 3/6) during the pandemic. Based on Castellini et al. [68], these effects were more consistent for the comparison with pre-pandemic data further back in time (January–September 2019) than for data from shortly before the pandemic outbreak (November 2019–January 2020). As indicated by vote counting, there was no clear difference in mental health changes depending on diagnoses of eating disorder (anorexia nervosa vs. bulimia nervosa) in Castellini et al. [68]. On the other hand, some effect directions—most of them found for patients with a history of binge eating disorder [51]—were classified as (un-)clear effects, (potentially) indicating higher peri-pandemic levels of depressive symptoms (↓: 1/2; ↘: 1/2), eating disorder-specific psychopathology (↓: 2/7; ↘: 1/7), and loss of control over eating behavior (↓: 1/4; ↘: 1/4). For peri-pandemic changes, the effects in most available comparisons (based on [86]) were classified as unclear effects (↗), potentially suggesting an improvement of mental health in the pandemic timeline until June 2020 concerning anxiety (1/1), depressive symptoms (1/1), stress (1/1), and well-being (1/1). The same applied to psychological distress (↗: 3/3), with Daly and Robinson [58] finding two unclear effects, potentially indicating reduced distress in April (vs. March 2020) and July (vs. April 2020), respectively. Findings for loss of control regarding overeating behavior were mixed, with one unclear effect (↘), potentially suggesting increased levels of this disorder-specific outcome later in the pandemic (i.e., after ‘lockdown’ measures in Belgium), while one clear effect indicated an improvement during the pandemic (↑; April–June 2020).

#### 3.3.9. Substance-Related and Addictive Disorders

For individuals with SUD, the evidence base for pre- to peri-pandemic changes indicated better pre-pandemic mental health, while effect directions were heterogeneous for peri-pandemic changes (Table 3). Most of the observed effects for pre-to peri-pandemic comparisons (only available from [87]) were classified as clear effects (↓) indicating a worsening of mental symptoms, specifically regarding alcohol use (1/3), anxiety (1/2) and depressive symptoms (1/2), stress (1/2), and tobacco use (2/2). For peri-pandemic changes of anxiety, depressive symptoms, and stress, based on Liu et al. [87], effect directions suggested an improvement of mental health in the pandemic course (↑: 1/1, respectively). Regarding substance use, there were clear and unclear effects, (potentially) indicating a decrease in use in the pandemic timeline for some substances (↑: alcohol: 1/3, ↗: alcohol: 1/3, cannabis: 2/2). On the other hand, further (un-)clear effects (potentially) suggested an increase in use for other substances during the pandemic (↓: amphetamine use: 1/1, morphine use: 1/1; ↘: heroin use: 1/1). Findings were mixed concerning the use of other drugs. As indicated by Table 3, the evidence on pandemic-related mental health changes in patients with gambling disorders was very limited. Only Donati et al. [80] examined peri-pandemic changes of gambling problem symptoms amid the lockdown implemented in Italy, with one clear effect (↑) indicating an improvement during the lockdown.

#### 3.3.10. Mixed Group

For individuals with various diagnoses of mental illness, a large variety of effect directions was observed for pre- to peri-pandemic changes (Table 3), including (potentially) better peri-pandemic mental health, but also deterioration during the pandemic and mixed findings. Regarding peri-pandemic changes, most observed effects suggested a potential improvement of mental health in the pandemic timeline. Apart from outcomes for which vote counting was not fully applicable, most observed effects for pre- to peri-pandemic comparisons of anxiety (↑: 2/5; ↗: 1/5), depressive symptoms (↑: 2/9; ↗: 1/9), psychological distress (↑: 2/6; ↗: 2/6), and stress (↑: 2/4; ↗: 1/4) were classified as (un-)clear effects, (potentially) suggesting a reduction of symptoms. Remaining effect directions found for these outcomes, mostly in patients with chronic disorder [54], were classified as unclear effects (↘), potentially indicating better pre-pandemic mental health (anxiety: 2/5; depressive symptoms: 2/9; psychological distress: 1/6; stress: 1/4). Findings for social functioning/relationships were mixed (↑: 1/5; ↗: 1/5; ∎: 1/5; ↘: 2/5). For well-being, one unclear effect based on Johnco et al. [63] potentially suggested better pre-pandemic values. Pinkham et al. [65] was the only study clustering mental health data of patients with bipolar disorder I or II (with/without psychotic features) and depressive disorder (with/without psychotic features) under the term affective disorders. For depressive symptoms, a null effect was observed. Otherwise, findings were mixed, with one clear (↑; well-being) and unclear effect (↗; sleep quantity) possibly indicating better peri-pandemic mental health. Regarding peri-pandemic changes, most effects’ directions were unclear, potentially indicating an improvement during the pandemic for depressive symptoms (↗: 4/5) and psychological distress (↗: 3/4), as well as a smaller impact of the pandemic on quality of life (↗: 1/1) and social functioning/relationships (↑: 1/3; ↗: 2/3). Findings were mixed for anxiety (↗: 1/4; ∎: 1/4; ↘: 1/4) and stress (↗: 1/2; ↘: 1/2).

## 4. Discussion

In this systematic review, we assessed the mental health impact of the COVID-19 pandemic in patients with pre-existing mental illness. We identified 40 studies measuring various mental health outcomes longitudinally or repeatedly cross-sectionally before and after the pandemic outbreak, across peri-pandemic time points, or both. Studies were conducted in individuals with ASD, schizophrenia spectrum and other psychotic disorders, bipolar disorders, depressive disorders, anxiety disorders, OCD, PTSD, eating disorders, and substance-related and addictive (gambling) disorders. Several mixed studies were conducted in patients with various diagnoses of mental illness. Overall, the evidence base provided heterogeneous findings on mental health consequences of the pandemic in individuals with pre-existing mental illness, both within and across diagnostic groups as well as for pre- to peri-pandemic and peri-pandemic comparisons. Based on narrative synthesis, the evidence seems to be more mixed than for the general population for which resilience (i.e., stable good mental health) was an often-observed trajectory of mental health amid the pandemic [89,90,91,92]. Few studies investigated differential mental health effects within a diagnostic group (e.g., MDD and persistent depressive disorder within depressive disorders [64,68]). Furthermore, potential variations depending on the type of symptoms [71] or stage of mental illness [54,73] were hardly considered.

Pre- to peri-pandemic changes of mental health. The evidence base suggested a (potential) improvement of mental health during the pandemic compared to before for patients with depressive disorders (especially persistent depressive disorder), anxiety disorders (especially GAD, social anxiety disorder), and eating disorders. Studies including patients with various mental diagnoses also observed effect directions, (possibly) suggesting an improvement amid the pandemic for many mental health outcomes. Especially for patients with stress-related mental illness such as depressive and anxiety disorder, better mental health early in the pandemic with its related measures of containment might at least partly be explained by an initial decrease in and/or elimination of demanding situations in daily life. For example, despite the pandemic as a more or less synchronously starting global macrostressor, exposure to daily hassles was found to decrease [93]. Given the reduced need for real-life social interactions and for venturing into public places or situations, especially individuals with pre-existing agoraphobia and social anxiety disorder could be less-burdened amid the pandemic. However, this observation based on effect directions was only made for individuals with social phobia, while findings for agoraphobia (and panic disorder) were heterogeneous. The mixed findings for anxiety disorders in this review might be explained by an increased risk of these patients to respond with pandemic- and COVID-19-related fears, as previously discussed [94]. Moreover, constructs such as intolerance of uncertainty, which are related to anxiety disorders [95], were found as predictors of COVID-19-related fears [96]. Thus, mental distress early in the pandemic might not have increased in this group, but fears regarding COVID-19 and (long-term) consequences of the pandemic could nevertheless lead to mixed findings. Based on the available evidence, there was no indication of a difference in mental health responses between individuals with anorexia and bulimia nervosa. The positive findings for these groups could simply be due to the early pandemic phase, when stay-at-home orders were implemented internationally for the first time. However, in the further pandemic course (i.e., beyond first wave and summer 2020 covered by this review) and with more frequent ‘lockdown’ periods, dysfunctional thoughts about weight could become more apparent, with negative consequences for eating behavior and mental distress.

According to the narrative synthesis, people with a diagnosis of MDD seemed to be negatively affected, with possibly poorer mental health since the pandemic outbreak. While the pandemic was associated with a decrease in microstressors [93], it did also negatively affected patients’ daily structure and expose them to other stressors, e.g., no group activities and mask wearing [97]. The loss of positive experiences and activities potentially led to poorer mental health in acutely depressive patients, with an increase in symptoms such as sadness, loss of energy, and lack of interest. Individuals with persistent depressive disorder might have been better able to compensate for this loss, as they were already supplied with mental health services at the pandemic onset. On the other hand, this result was not consistently found across the included studies [54]. As mentioned above, the (potentially) poorer peri-pandemic mental health in individuals with binge eating disorder in this review might be due to an attentional focus toward food early in the pandemic in preparation for stay-at-home restrictions [22]. Poorer pre-pandemic mental health was also identified for individuals with SUD, potentially due to lack of social support (e.g., peer support groups) during stay-at-home orders and limited access to clinical treatment [17], which could not yet be compensated by telemedical or other types of support early in the pandemic.

Regarding pre- to peri-pandemic changes in mental health, the unavailable or very limited evidence hindered clear conclusions for individuals diagnosed with ASD, anxiety disorders, PTSD, and substance-related and addictive disorders, although data on bipolar and depressive disorders were also restricted to two studies. No clear conclusions were also possible for people with schizophrenia spectrum or other psychotic disorders and bipolar disorders due to mixed results in this review, which might be explained by varying pandemic situations (e.g., COVID-19 incidence) in different countries and heterogeneous samples, e.g., chronic vs. acute schizophrenia vs. patients with psychosis [57,65,69,71].

Peri-pandemic changes of mental health. The narrative synthesis indicated (potential) psychological adaptation in the pandemic timeline for several diagnostic groups. For example, in individuals diagnosed with ASD, psychological distress increased compared to before the pandemic based on an individual study. However, the proportion of effect directions for peri-pandemic comparisons suggested that their mental health status may have recovered between an early phase of the pandemic (mostly spring 2020) and later time points (until May 2020). Similar changes were also observed for other groups, although only based on Daly and Robinson [58] for psychological distress. After an initial deterioration early in the pandemic, effect directions indicated a (potential) recovery for individuals with pre-existing diagnoses of depressive and anxiety disorders, OCD, and PTSD. These narrative findings are in line with previous evidence on pandemic-related trajectories of mental health in the general population showing a recovery of mental health after temporarily increased mental dysfunctions [89,92,98].

Probably coinciding with and related to globally relaxed measures of containment during the summer period 2020, our review also found an improvement of mental health across peri-pandemic assessments for individuals with pre-existing eating disorders and gambling disorders. The decrease in gambling problem symptoms might possibly be due to limited possibilities of (pathological) gambling during public health measures (e.g., business closures), which might not have been lifted everywhere until summer 2020. However, data are restricted to the early pandemic phase, limiting these preliminary positive findings.

The narrative synthesis for peri-pandemic changes also allowed no clear conclusions based on the limited available evidence (depressive disorders, anxiety disorders, OCD, PTSD, gambling disorders) or mixed evidence (schizophrenia, bipolar disorders, SUD) for several diagnoses. Mixed results might also be due to dynamic pandemic situations (e.g., different speed of countries in lifting containment measures during summer 2020) and heterogeneous sampling constellations (e.g., bipolar disorder types I and II).

Overall, our narrative findings on partly negative mental health effects of the pandemic replicate the results of some systematic reviews [4,25], with this review being able to include more longitudinal studies providing comparative data. Our observation of (potentially) better mental health compared to a pre-pandemic baseline for some groups goes beyond the finding of Robinson et al. [32], who showed no evidence of any change in symptoms among samples with pre-existing mental health conditions. Given the lack of longitudinal studies identified by previous reviews, they did not make any conclusions regarding peri-pandemic changes of mental health, rendering comparisons difficult. For specific diagnoses of mental illness, our more heterogeneous findings partly contradict previous evidence syntheses that mostly concluded a worsening of symptoms in individuals with eating [35,36] and bipolar disorders [37], for example, although these reviews did also hardly identify any longitudinal research.

Limitations. We conducted searches across four (COVID-19-specific) electronic databases, without temporal or geographical restrictions. Although we used no language restrictions, it is possible that we missed relevant non-English language studies. The search strategy was restricted to databases, without considering additional sources such as reference lists. Thus, it is possible that some relevant studies were not identified. We did not assess the certainty of evidence and were unable to conduct meta-analyses. However, we relied on the most up-to date narrative and graphical summary approach to meaningfully summarize findings and to reduce bias in reporting. To consider both the direction of effects and statistical significance, vote counting using effect direction was adapted for this review. On the other hand, vote counting is also limited, as it does not provide information on the magnitude of effects or account for differences in the relative study sizes [50]. Some included studies providing several comparisons were considered multiple times for vote counting to make clearer conclusions regarding pandemic-related mental health impacts (e.g., whether mental health changed differently in patients with various diagnoses or in studies providing high-frequent assessment) and to derive more specific implications. The focus of our review on diagnoses of mental disorders might also have been a limitation and could have contributed to the heterogeneity in our findings, as diagnoses per se are heterogeneous in nature. Based on a sufficient number of primary studies measuring these outcomes, future systematic reviews might also investigate the effects of the pandemic on symptom dimensions (e.g., cognitive domains, positive and negative affect domains) to take into account potential diverging effects of the pandemic on specific mental constructs rather than diagnoses. However, this was not possible based on the evidence found in this review (see Appendix A). The included studies also had some limitations. The exposure to the pandemic and containment measures as geographically and temporally often varying stressors even differed within studies, possibly also heterogeneously affecting mental health in individuals with pre-existing mental disorders. Sample sizes were often small, and most studies did not control for potential confounding variables (e.g., severity of containment measures). The available evidence was limited regarding geographical location (i.e., mostly Western, high-income countries), gender (i.e., mostly females), and age distribution (i.e., mostly young-to-middle-aged adults). The paucity of medium- or long-term data beyond the early phase also restricts the generalizability to later time points in the pandemic timeline. Finally, due to the primary use of self-report scales, no conclusions can be made about whether the prevalence of diagnoses did change amid the pandemic (e.g., new diagnosis of comorbid mental disorders). Although this review considered a broad range of mental health outcomes, positive outcomes such as well-being were hardly measured, rendering conclusions about potential beneficial psychological effects of the pandemic difficult.

Implications for research. Based on a lack of studies in non-Western and low- and middle-income countries, research in these regions is urgently needed. Since being at particular risk for severe courses of COVID-19 [99], which might also negatively affect their mental health, older adults with pre-existing mental illness should be focused on more. Furthermore, the limited body of evidence indicates the compelling need for further research on several diagnostic groups. For example, symptoms of depressive and anxiety disorder as well as PTSD, that is, mental disorders whose onset and persistence is often related to stressors [100,101], might have been exacerbated by pandemic stressors, although we identified only few studies in these groups. Individuals with pre-existing bipolar disorder were comparatively well-represented in the evidence base, but findings of narrative synthesis were mixed, indicating the need for forthcoming research to clarify this heterogeneity. Since they were not the focus of this review, mental health effects in individuals with personality disorder should be studied. Furthermore, this review did not aim at health services research and outcomes such as mental healthcare utilization/access. However, in view of pandemic-related disruptions of mental health services [8,39], both mental health consequences and healthcare supply of individuals with mental illness in view of the pandemic and future major disruptive events should be investigated.

Study designs need to be improved by examining larger sample sizes and controlling for potential confounding variables (e.g., COVID-19 incidence rates, severity of containment measures, and chronicity of mental illness). Although the pandemic presents a major global disruption that is persistent in nature and affecting people with mental health conditions worldwide, individual stressor exposure might have considerably differed, both concerning pandemic-related stressors and additionally occurring (micro-/macro-)stressors. Further longitudinal research and respective reviews are needed to assess medium- and long-term psychological consequences of the COVID-19 pandemic. While mental health, due to the initial elimination of stressful situations amid measures of containment, might have improved early in the pandemic in some diagnostic groups (e.g., social anxiety disorder, ASD), the increasing lack of exposure to disease-promoting factors and stressors (e.g., social situations) might reinforce avoidance tendencies and lead to more mental distress in the long-term.

Observational studies should take a more differentiated view by examining individuals with different diagnoses of mental illness to make specific conclusions and recommendations for interventions (e.g., psychotherapy). Comparable to the general population, it would be worthwhile to examine (longitudinal) trajectories of mental distress to identify distinct patterns of mental response to the pandemic and dynamic changes more clearly, both across and within diagnostic groups. The study of disorder-specific and transdiagnostic predictors of these trajectories would help to derive clinical implications for individuals with specific diagnoses and to offer overarching mental health services.

Finally, the included studies primarily focused on individuals with chronic mental illness who might have been less negatively affected as they were already provided with psychotherapeutic and/or psychiatric services. Patients diagnosed shortly before the pandemic outbreak or with acute symptoms, however, could have been more at risk for poorer mental health, as they were not yet integrated into mental healthcare structures, which were further disrupted by the pandemic. Therefore, especially in view of likely future major disruptions that will affect societies at large (e.g., pandemics, critical events related to climate change, wars), research should also focus on individuals with on-setting mental illness.

Implications for practice. In view of mixed findings and the limited evidence base for many diagnostic groups, we can only derive limited implications for practice. From a general point of view, it seems worthwhile to consider variations in mental health responses between different diagnoses of mental illness. For example, our narrative synthesis provided preliminary evidence that individuals with various diagnoses of anxiety disorder were affected heterogeneously by the pandemic. Similarly, more psychological support services might be needed depending on the level of chronicity of disease, although the findings of vote counting were heterogeneous in this respect. Provided that more longitudinal research is conducted to replicate these findings and to clarify contradictory results, mental health services during the pandemic might also have a larger impact if they were provided in a differentiated way.

This review observed various pre- to peri-pandemic as well as peri-pandemic changes of mental health in people with pre-existing mental illness. Thus, mental healthcare might benefit from frequent monitoring of these patients’ mental health in presence of a major stressor to be able to respond more quickly to short-term increases in psychological distress. The availability of need-oriented mental health services that are tailored to the respective patient subgroup and provided dynamically depending on external stressors (e.g., containment measures), could also be important in view of possible future global disruptive events.

## 5. Conclusions

The present systematic review including a narrative synthesis of longitudinal observational studies suggests heterogeneous mental health consequences of the COVID-19 pandemic in people with pre-existing mental illness, partly depending on the underlying diagnosis. The body of evidence, which is limited to the early pandemic phase, high-income countries, and young-to-middle-aged female individuals with chronic mental disorders, indicated (potentially) poorer as well as better peri-pandemic mental health compared to pre-pandemic data. For peri-pandemic changes, the available evidence was more limited, with indications of psychological adaptation but also mixed findings. The review highlights the compelling need for well-designed studies, specifically measuring the impact of the pandemic in the medium- and long-term. Future research should focus on thus far little-studied diagnostic groups, considering diagnoses in a differentiated way, and control for the severity of public health measures to contain the pandemic. These findings might improve mental health services amid the ongoing pandemic as well as help to face upcoming major disruptions.

## Figures and Tables

**Figure 1 ijerph-20-00948-f001:**
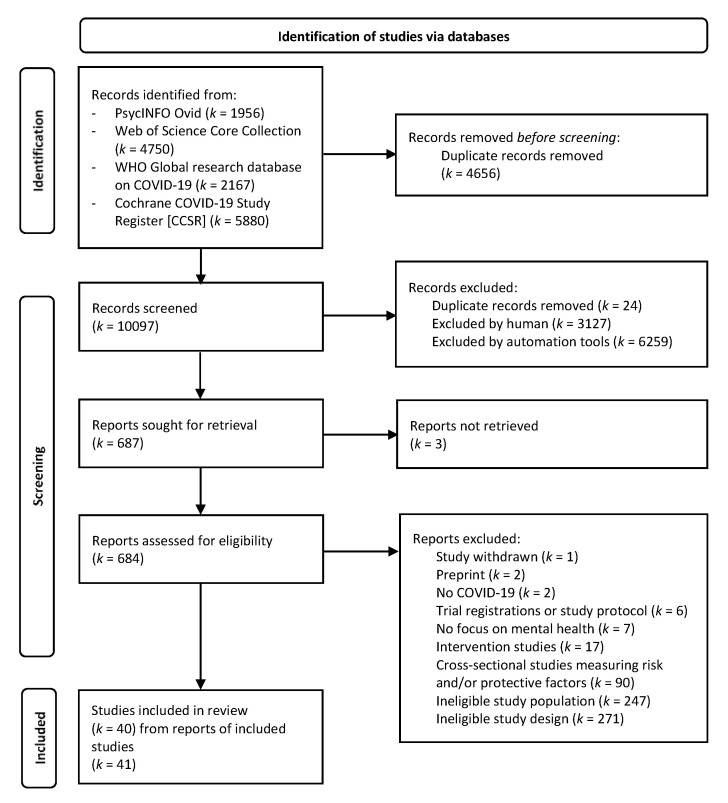
Flow chart of study selection process following PRISMA guidelines [46]. Note. *k* = number of studies.

**Figure 2 ijerph-20-00948-f002:**
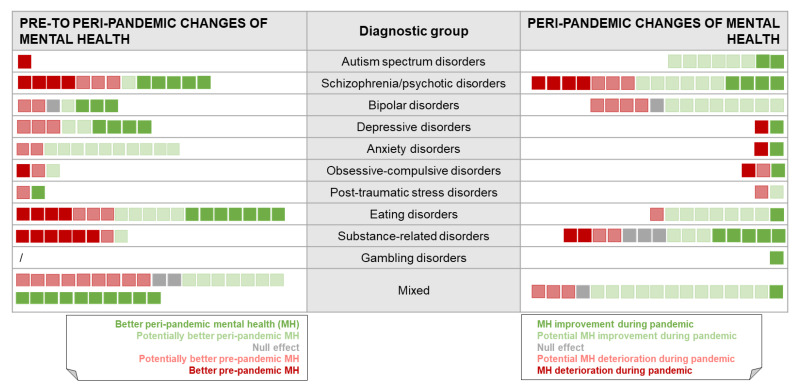
Graphical summary of results across diagnostic groups (*k* = 40 observational studies).

**Table 1 ijerph-20-00948-t001:** Vote counting based on direction of effects.

**Pre- to peri-pandemic changes** For each study providing mental health data prior to and after the pandemic outbreak, mental health effects were classified as one of the following five effect directions, based on the available comparisons for a specific outcome.	
Direction 1: Clear effect for better peri-pandemic MH (↑)	Effect in the direction of pandemic assessment (i.e., mental health improved during the pandemic compared to before), with clear indication that the 95% confidence interval (CI) does not contain the null or *p* < 0.05.
Direction 2: Unclear effect for potentially better peri-pandemic MH (↗)	Effect in the direction of pandemic assessment (i.e., mental health improved compared to before); however, either the 95% CI contains the null effect or *p* > 0.05.
Direction 3: Null effect (∎)	No difference seen between peri-pandemic and pre-pandemic assessment. We classified studies only narratively reporting that ‘no significant difference was observed’ (without further details) as well as studies reporting an actual null effect based on an effect estimate (e.g., Cohen’s *d* = 0).
Direction 4: Unclear effect for potentially better pre-pandemic MH (↘)	Effect in the direction of pre-pandemic assessment (i.e., better mental health before the pandemic); however, either the 95% CI contains the null effect or *p* > 0.05.
Direction 5: Clear effect for better pre-pandemic MH (↓)	Effect in the direction of pre-pandemic assessment (i.e., better mental health before the pandemic), with clear indication that the 95% CI does not contain the null or *p* < 0.05.
**Peri-pandemic changes** For each study reporting several peri-pandemic assessments, changes in mental health were classified as one of the following five effect directions, based on the available comparisons for a specific outcome.	
Direction 1: Clear effect for MH improvement during the pandemic (↑)	Effect in the direction of later pandemic assessment (i.e., mental health improved over the course of the pandemic), with clear indication that the 95% confidence interval (CI) does not contain the null or *p* < 0.05.
Direction 2: Unclear effect for potential MH improvement during the pandemic (↗)	Effect in the direction of later pandemic assessment (i.e., mental health improved over the course of the pandemic); however, either the 95% CI contains the null effect or *p* > 0.05.
Direction 3: Null effect (∎)	No difference seen between peri-pandemic assessments. We classified studies only narratively reporting that ‘no significant difference was observed’ (without further details) as well as studies reporting an actual null effect based on an effect estimate (e.g., Cohen’s *d* = 0).
Direction 4: Unclear effect for potential MH deterioration during the pandemic (↘)	Effect in the direction of earlier pandemic assessment (i.e., mental health deteriorated over the course of the pandemic); however, either the 95% CI contains the null effect or *p* > 0.05.
Direction 5: Clear effect for MH deterioration during the pandemic (↓)	Effect in the direction of earlier pandemic assessment (i.e., mental health deteriorated over the course of the pandemic), with clear indication that the 95% CI does not contain the null or *p* < 0.05.

Note. MH: mental health; *p*: *p*-value.

**Table 3 ijerph-20-00948-t003:** Vote counting summary.

Outcomes	Clear Effect for Better Peri-Pandemic MH/MH Improvement during Pandemic	Unclear Effect for Potentially Better Peri-Pandemic MH/Potential MH Improvement during Pandemic	NA ^1^ (Better Peri-Pandemic MH/MH Improvement during Pandemic)	Null Effect	NA ^2^ (Better Pre-Pandemic MH/MH Deterioration during Pandemic)	Unclear Effect for Potentially Better Pre-Pandemic MH/Potential MH Deterioration during Pandemic	Clear Effect for Better Pre-Pandemic MH/MH Deterioration during Pandemic
**A. Pre- to peri-pandemic changes of mental health**							
**Autism spectrum disorders**							
Psychological distress	-	-	-	-	-	-	↓
**Schizophrenia spectrum and other psychotic disorders**							
Anxiety	-	-	-	-	-	-	↓
Depressive symptoms	-	-	-	-	-	↘	-
Diminished motivation and pleasure	-	-	-	-	-	-	↓
Excitement/energization	↑↑	-	-	-	-	-	↓
Hallucination	↑	↗	-	-	-	-	↓
Sleep quantity	-	-	-	-	-	↘	-
Substance use	↑	-	-	-	-	↘	-
Well-being	↑	-	-	-	-	-	-
**Bipolar disorders**							
Anxiety	↑	-	-	-	-	↘	-
Depressive symptoms	↑	-	-	-	-	↘	-
(Hypo-)manic symptoms	↑	-	-	-	-	-	-
Loneliness	-	-	-	∎	-	-	-
Sleep quality ^3^	-	↗	-	-	-	-	-
**Depressive disorders**							
Anxiety	↑	↗	-	-	-	↘	-
Depressive symptoms	↑↑	↗	-	-	-	↘	-
Loneliness	↑	-	-	-	-	↘	-
**Anxiety disorders**							
Anxiety	-	↗↗↗	-	-	-	↘	-
Depressive symptoms	-	↗↗↗↗	-	-	-	-	-
Loneliness	-	↗↗↗	-	-	-	↘	-
**Obsessive-compulsive disorders ^4^**							
Anxiety	-	-	-	-	-	↘	-
Loneliness	-	↗	-	-	-	-	-
OCD severity	-	-	△_NA_	-	▽_NA_▽_NA_	-	↓
**Post-traumatic stress disorders**							
Depressive symptoms	-	-	-	-	-	↘	-
Post-traumatic stress symptoms	↑	-	-	-	-	-	-
**Eating disorders**							
Depressive symptoms	-	-	-	-	-	↘	↓
Eating disorder-specific psychopathology	↑↑↑	↗	-	-	-	↘	↓ ↓
Loss of control overeating behavior ^5^	↑	↗	-	-	-	↘	↓
Psychological distress	↑↑↑	↗↗↗	-	-	-	-	-
**Substance-related disorders**							
Alcohol use	-	-	-	-	▽_NA_	↘	↓
Amphetamine use	-	-	△_NA_	-	▽_NA_	-	-
Anxiety	-	↗	-	-	-	-	↓
Depressive symptoms	-	-	-	-	▽_NA_	-	↓
Morphine use	-	-	△_NA_	-	▽_NA_	-	-
Stress	-	-	-	-	▽_NA_	-	↓
Tobacco use	-	-	-	-	-	-	↓↓
**Gambling disorders**							
-	-	-	-	-	-	-	-
**Mixed (no subgroup data reported)**							
Anxiety	↑↑	↗	-	-	-	↘↘	-
Depressive symptoms	↑↑	↗	△_NA_ △_NA_△_NA_	∎	▽_NA_	↘↘	-
Loneliness (thwarted belongingness)	-	-	△_NA_ △_NA_	-	▽_NA_ ▽_NA_	-	-
Psychological distress	↑↑	↗↗	-	-	▽_NA_	↘	-
Sleep quantity	-	↗	-	-	-	-	-
Social functioning/relationships	↑	↗	-	∎	-	↘↘	-
Stress	↑↑	↗	-	-	-	↘	-
Suicidality	-	-	△_NA_ △_NA_ △_NA_ △_NA_	-	-	-	-
Well-being ^6^	↑	-	-	-	-	↘	-
**B. Peri-pandemic changes of mental health**							
**Autism spectrum disorders**							
Anxiety	-	↗↗	-	-	-	-	-
Depressive symptoms	↑	↗	-	-	-	-	-
Psychological distress	↑	↗↗	-	-	-	-	-
Stress	-	↗	△_NA_	-	-	-	-
**Schizophrenia spectrum and other psychotic disorders**							
Anxiety	↑	-	-	-	-	-	↓
Depressive symptoms	↑	-	-	-	-	↘	-
Diminished motivation and pleasure	-	-	-	-	-	↘	-
Loneliness	↑	-	-	-	-	-	-
Psychological distress	-	↗↗↗↗	-	-	-	↘	-
Sleep quality	-	-	-	-	-	-	↓
Stress	↑	-	-	-	-	-	↓↓
Substance use	-	↗↗	-	-	-	-	-
**Bipolar disorders**							
Anxiety	-	↗↗	-	-	-	-	-
Circadian rhythms	-	-	-	-	-	↘	-
Depressive symptoms	-	↗↗↗	-	∎	-	-	-
Psychological distress	-	-	-	-	-	↘↘	-
Sleep quality ^3^	-	↗↗	-	-	-	-	-
Stress ^7^	-	↗	-	-	-	↘	-
**Depressive disorders**							
Psychological distress	↑	-	-	-	-	-	↓
**Anxiety disorders**							
Psychological distress	↑	-	-	-	-	-	↓
**Obsessive-compulsive disorders**							
Psychological distress	↑	-	-	-	-	↘	-
OCD severity	-	-	-	-	-	-	↓
**Post-traumatic stress disorders**							
Psychological distress	-	↗	-	-	-	↘	-
**Eating disorders**							
Anxiety	-	↗	-	-	-	-	-
Depressive symptoms	-	↗	-	-	-	-	-
Loss of control overeating behavior ^5^	↑	-	-	-	-	↘	-
Psychological distress	-	↗↗↗	-	-	-	-	-
Stress	-	↗	-	-	-	-	-
Well-being	-	↗	-	-	-	-	-
**Substance-related disorders**							
Alcohol use	↑	↗	-	∎	-	-	-
Amphetamine use	-	-	-	-	-	-	↓
Anxiety	↑	-	-	-	-	-	-
Cannabis/marijuana use	-	↗↗	-	-	-	-	-
Cocaine use	-	-	△_NA_	-	-	-	-
Depressive symptoms	↑	-	-	-	-	-	-
Heroin use	-	-	-	-	-	↘	-
Morphine use	-	-	-	-	-	-	↓
Psychological distress	-	-	-	∎	-	-	-
Stress	↑	-	-	-	-	-	-
Tobacco use	-	-	-	-	▽_NA_	-	-
Use of other drugs	↑	-	-	-	-	↘	-
Use of prescription drugs	-	-	-	∎	-	-	-
**Gambling disorders**							
Gambling problem symptoms	↑	-	-	-	-	-	-
**Mixed (no subgroup data reported)**							
Anxiety	-	↗	△_NA_	∎	-	↘	-
Depressive symptoms	-	↗↗↗↗	-	-	▽_NA_	-	-
Loneliness (thwarted belongingness)	-	-	△_NA_	-	-	-	-
Psychological distress	-	↗↗↗	-	-	-	↘	-
Social functioning/relationships	↑	↗↗	-	-	-	-	-
Stress	-	↗	-	-	-	↘	-
Suicidality	-	-	△_NA_	-	▽_NA_	-	-
Well-being ^6^	-	↗	-	-	-	-	-

Note. Diagnostic groups ordered by DSM-5. Symbols represent the effect direction: ↑(direction 1): clear effect for better peri-pandemic mental health (MH) compared to pre-pandemic situation or MH improvement during the pandemic (i.e., better MH at later vs. earlier peri-pandemic assessment); ↗ (direction 2): unclear effect for potentially better peri-pandemic MH (vs. pre-pandemic situation) or potential MH improvement during the pandemic (i.e., better MH at later vs. earlier peri-pandemic assessment); ∎ (direction 3): null effect, studies only narratively reporting that ‘no significant difference was observed’, studies reporting an actual null effect based on an effect estimate (e.g., Cohen’s *d* = 0); ↘ (direction 4): unclear effect for potentially better pre-pandemic MH (vs. peri-pandemic situation) or potential MH deterioration during the pandemic (i.e., better MH at earlier vs. later peri-pandemic assessment); ↓ (direction 5): clear effect for better pre-pandemic MH (vs. peri-pandemic situation) or MH deterioration during the pandemic (i.e., better MH at earlier vs. later peri-pandemic assessment). *Abbreviations:* MH: mental health; NA: not applicable; OCD severity: severity of obsessive-compulsive symptoms. ^1^ △_NA_: Based on reported statistical values, probably clear or unclear effect for (potentially) better peri-pandemic MH/(potential) MH improvement during the pandemic. However, vote counting was not possible since test and/or *p*-value were not reported to assign to direction 1 or 2. ^2^ ▽_NA_: Based on reported statistical values, probably clear or unclear effect for (potentially) better pre-pandemic MH/(potential) MH deterioration during the pandemic. However, vote counting was not possible since test and/or *p*-value were not reported to assign to direction 4 or 5. ^3^ Bad sleep quality. ^4^ Chakraborty and Karmakar [23]: this study measured OCD severity but did not report a statistical analysis and no category of effect direction could be determined. ^5^ Including binge eating episodes. ^6^ Including quality of life. ^7^ Including pandemic-related stress.

## Data Availability

The data presented in this review are available in the article and its Appendix A.

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
