# Peer review of "Mental Health Impact of Early Stages of the COVID-19 Pandemic on Individuals with Pre-Existing Mental Disorders: A Systematic Review of Longitudinal Research"

_ijerph, 2023, doi:10.3390/ijerph20020948_

Round 1

Reviewer 1 Report

Very well put together comprehensive study

1. What is the main question addressed by the research? What is the impact of the Pandemic on individuals with pre-existing mental health disorders
2. Do you consider the topic original or relevant in the field? Does it
address a specific gap in the field?   This is very original and important and there has been much talk and speculation about this question without good data to back up opinions. This paper provides a nice balanced and very done study reviewing this topic.  
3. What does it add to the subject area compared with other published
material?   This is a very well done systematic review. 
4. What specific improvements should the authors consider regarding the
methodology? What further controls should be considered?   I suppose that the effect of mental health on those without pre-existing mental disorders would be good, but the paper is pretty large and comprehensive already. 
5. Are the conclusions consistent with the evidence and arguments presented
and do they address the main question posed?   Yes, the data is nicely synthesized and summarized.
6. Are the references appropriate?   Very comprehensive. 
7. Please include any additional comments on the tables and figures.   The tables are very comprehensive but the authors might consider putting some of the larger tables as supplemental material to make the paper more compact.

Reviewer 2 Report

In this review authors offer a good synthesis of the general effects of COVID19 pandemic in individuals pre-existing mental illness. In details, they collected evidences in paper focused on the effects of the pandemic on several psychiatric groups (including but not limited to Schizophrenia (scz), Bipolar (bd) and Depressive disorders (mdd)) and focused their attention on pre-pandemic/peri-pendemic changes and changes within the pandemic. As the authors described in their work, the pandemic and the related forced isolation had heterogeneous effects on the psychiatric diagnoses, in particular for bd scz and mdd.

Authors discussed potential reasons and limits (eg. selected studies relatively small sample size) to explain this result. If may add a comment, another possible reason could be related to the choice of focusing on diagnoses (as diagnoses themselves are heterogenic in nature). In this sense, I think it would have been interesting to see the effects of the pandemic on single symptoms' dimensions (eg. cognitive domains, positive and negative valence domains, etc..) in order to take into account potential diverging effects of covid pandemic on specific constructs rather than diagnoses. But to my knowledge there is hardly any work that took this approach so a review on that may not have been possible.  

Content-wise i have seen no major issues, nor i have particular suggestions to make. 

I only point to one minor point to address.

- lane 183-184: titles/abstracts were screened by two reviewers working independently (AMK, CK, MC, NR, RME, SL)

two reviewers are indicated but there are six authors. A similar issue can be found in lane 190-191

In my opinion, the paper can be accepted in the present form for publication. I thank the authors for their work.

Reviewer 3 Report

Dear Authors,

The manuscript entitled “Mental health impact of early stages of the COVID-19 pandemic on individuals with pre-existing mental disorders: A systematic review of longitudinal research” is devoted to analyses of published studies about changes in mental health during the pandemic.  

This is a very current issue.

·       The title of the presented article corresponds with the main text.

·       The abstract is describing the properly about the discussed problem.

·       The introduction seems to be too detailed but I understand the intention which leads to an explanation of why the topic was chosen and then what is the aim.

·       The materials and methods are advanced

·       The results are very detailed and properly described.

·       Deusscusion is like a sum up of collected results. It is divided into a part which makes reading easier and more understandable, especially with so many results. Authors as well are writing down the limitations of their work and implications.

I think it is a very interesting and very valuable article. I have only very small suggestions:

·       The key words: it is unnecessary to put  “COVID-19, pandemics” because they are already in the title. I would suggest using for e.g. SARS-CoV2,  infection,

·       Line 46: Please use capital letters when you write about  „severe acute respiratory syndrome coronavirus 2”

·       Line 120; 123; 124 – Authors should write as it was earlier and later is  -  Robinson et al. [31] ;  Nellam et al. [4]; Fleischmann et al. [25]

In my opinion, this manuscript is worth publishing.

With highest regards,

Reviewer
